# Statistical Model Criticism
# using Kernel Two Sample Tests

**James Robert Lloyd**
Department of Engineering
University of Cambridge

**Zoubin Ghahramani**
Department of Engineering
University of Cambridge

## Abstract

We propose an exploratory approach to statistical model criticism using maximum mean discrepancy (MMD) two sample tests. Typical approaches to model criticism require a practitioner to select a statistic by which to measure discrepancies between data and a statistical model. MMD two sample tests are instead constructed as an analytic maximisation over a large space of possible statistics and therefore automatically select the statistic which most shows any discrepancy. We demonstrate on synthetic data that the selected statistic, called the witness function, can be used to identify where a statistical model most misrepresents the data it was trained on. We then apply the procedure to real data where the models being assessed are restricted Boltzmann machines, deep belief networks and Gaussian process regression and demonstrate the ways in which these models fail to capture the properties of the data they are trained on.

## 1 Introduction

Statistical model criticism or checking[1] is an important part of a complete statistical analysis. When one fits a linear model to a data set a complete analysis includes computing e.g. Cook's distances [3] to identify influential points or plotting residuals against fitted values to identify non-linearity or heteroscedasticity. Similarly, modern approaches to Bayesian statistics view model criticism as in important component of a cycle of model construction, inference and criticism [4].

As statistical models become more complex and diverse in response to the challenges of modern data sets there will be an increasing need for a greater range of model criticism procedures that are either automatic or widely applicable. This will be especially true as automatic modelling methods [e.g. 5, 6, 7] and probabilistic programming [e.g. 8, 9, 10, 11] mature.

Model criticism typically proceeds by choosing a statistic of interest, computing it on data and comparing this to a suitable null distribution. Ideally these statistics are chosen to assess the utility of the statistical model under consideration (see applied examples [e.g. 4]) but this can require considerable expertise on the part of the modeller. We propose an alternative to this manual approach by using a statistic defined as a supremum over a broad class of measures of discrepancy between two distributions, the maximum mean discrepancy (MMD) [e.g. 12]). The advantage of this approach is that the discrepancy measure attaining the supremum automatically identifies regions of the data which are most poorly represented by the statistical model fit to the data.

We demonstrate MMD model criticism on toy examples, restricted Boltzmann machines and deep belief networks trained on MNIST digits and Gaussian process regression models trained on several time series. Our proposed method identifies discrepancies between the data and fitted models that would not be apparent from predictive performance focused metrics. It is our belief that more effort should be expended on attempting to falsify models fitted to data, using model criticism techniques or otherwise. Not only would this aid research in targeting areas for improvement but it would give greater confidence in any conclusions drawn from a model.

## 2 Model criticism

Suppose we observe data $Y^{\text{obs}} = (y_i^{\text{obs}})_{i=1\ldots n}$ and we attempt to fit a model $M$ with parameters $\theta$. After performing a statistical analysis we will have either an estimate, $\hat{\theta}$, or an (approximate) posterior, $p(\theta \mid Y^{\text{obs}}, M)$, for the parameters. How can we check whether any aspects of the data were poorly modelled?

**Criticising prior assumptions**  The classical approach to model criticism is to attempt to falsify the null hypothesis that the data could have been generated by the model $M$ for some value of the parameters $\theta$ i.e. $Y^{\text{obs}} \sim p(Y \mid \theta, M)$. This is typically achieved by constructing a statistic $T$ of the data whose distribution does not depend on the parameters $\theta$ i.e. a pivotal quantity. The extent to which the observed data $Y^{\text{obs}}$ differs from expectations under the model $M$ can then be quantified with a tail-area based $p$-value

$$p_{\text{freq}}(Y^{\text{obs}}) = \mathbb{P}(T(Y) \geq T(Y^{\text{obs}})) \quad \text{where} \quad Y \sim p(Y \mid \theta, M) \quad \text{for any } \theta. \tag{2.1}$$

Analogous quantities in a Bayesian analysis are the prior predictive $p$-values of Box [1]. The null hypothesis is replaced with the claim that the data could have been generated from the prior predictive distribution $Y^{\text{obs}} \sim \int p(Y \mid \theta, M)p(\theta \mid M)\mathrm{d}\theta$. A tail-area $p$-value can then be constructed for any statistic $T$ of the data

$$p_{\text{prior}}(Y^{\text{obs}}) = \mathbb{P}(T(Y) \geq T(Y^{\text{obs}})) \quad \text{where} \quad Y \sim \int p(Y \mid \theta, M)p(\theta \mid M)\mathrm{d}\theta. \tag{2.2}$$

Both of these procedures construct a function of the data $p(Y^{\text{obs}})$ whose distribution under a suitable null hypothesis is uniform i.e. a $p$-value. The $p$-value quantifies how surprising it would be for the data $Y^{\text{obs}}$ to have been generated by the model. The different null hypotheses reflect the different uses of the word 'model' in frequentist and Bayesian analyses. A frequentist model is a class of probability distributions over data indexed by parameters whereas a Bayesian model is a joint probability distribution over data and parameters.

**Criticising estimated models or posterior distributions**  A contrasting method of Bayesian model criticism is the calculation of posterior predictive $p$-values $p_{\text{post}}$ [e.g. 13, 14] where the prior predictive distribution in (2.2) is replaced with the posterior predictive distribution $Y \sim \int p(Y \mid \theta, M)p(\theta \mid Y^{\text{obs}}, M)\mathrm{d}\theta$. The corresponding test for an analysis resulting in a point estimate of the parameters $\hat{\theta}$ would use the plug-in predictive distribution $Y \sim p(Y \mid \hat{\theta}, M)$ to form the plug-in $p$-value $p_{\text{plug}}$.

These $p$-values quantify how surprising the data $Y^{\text{obs}}$ is even after having observed it. A simple variant of this method of model criticism is to use held out data $Y^*$, generated from the same distribution as $Y^{\text{obs}}$, to compute a $p$-value i.e. $p(Y^*) = \mathbb{P}(T(Y) \geq T(Y^*))$. This quantifies how surprising the held out data is after having observed $Y^{\text{obs}}$.

**Which type of model criticism should be used?**  Different forms of model criticism are appropriate in different contexts, but we believe that posterior predictive and plug-in $p$-values will be most often useful for highly flexible models. For example, suppose one is fitting a deep belief network to data. Classical $p$-values would assume a null hypothesis that the data could have been generated from *some* deep belief network. Since the space of all possible deep belief networks is very large it will be difficult to ever falsify this hypothesis. A more interesting null hypothesis to test in this example is whether or not our *particular* deep belief network can faithfully mimic the distribution of the sample it was trained on. This is the null hypothesis of posterior or plug-in $p$-values.

## 3 Model criticism using maximum mean discrepancy two sample tests

We assume that our data $Y^{\text{obs}}$ are i.i.d. samples from some distribution $(y_i^{\text{obs}})_{i=1\ldots n} \sim_{\text{iid}} p(y \mid \theta, M)$. After performing inference resulting in a point estimate of the parameters $\hat{\theta}$, the null hypothesis associated with a plug-in $p$-value is $(y_i^{\text{obs}})_{i=1\ldots n} \sim_{\text{iid}} p(y \mid \hat{\theta}, M)$.

We can test this null hypothesis using a two sample test [e.g. 15, 16]. In particular, we have samples of data $(y_i^{\text{obs}})_{i=1\ldots n}$ and we can generate samples from the plug-in predictive distribution $(y_i^{\text{rep}})_{i=1\ldots m} \sim_{\text{iid}} p(y \mid \hat{\theta}, M)$ and then test whether or not these samples could have been generated

from the same distribution. For consistency with two sample testing literature we now switch notation; suppose we have samples $X = (x_i)_{i=1...m}$ and $Y = (y_i)_{i=1...n}$ drawn i.i.d. from distributions $p$ and $q$ respectively. The two sample problem asks if $p = q$.

A way of answering the two sample problem is to consider maximum mean discrepancy (MMD) [e.g. 12] statistics

$$\text{MMD}(\mathcal{F}, p, q) = \sup_{f \in \mathcal{F}} (\mathbb{E}_{x \sim p}[f(x)] - \mathbb{E}_{y \sim q}[f(y)]) \tag{3.1}$$

where $\mathcal{F}$ is a set of functions. When $\mathcal{F}$ is a reproducing kernel Hilbert space (RKHS) the function attaining the supremum can be derived analytically and is called the witness function

$$f(x) = \mathbb{E}_{x' \sim p}[k(x, x')] - \mathbb{E}_{x' \sim q}[k(x, x')] \tag{3.2}$$

where $k$ is the kernel of the RKHS. Substituting (3.2) into (3.1) and squaring yields

$$\text{MMD}^2(\mathcal{F}, p, q) = \mathbb{E}_{x, x' \sim p}[k(x, x')] + 2\mathbb{E}_{x \sim p, y \sim q}[k(x, y)] + \mathbb{E}_{y, y' \sim q}[k(y, y')]. \tag{3.3}$$

This expression only involves expectations of the kernel $k$ which can be estimated empirically by

$$\text{MMD}_b^2(\mathcal{F}, X, Y) = \frac{1}{m^2} \sum_{i,j=1}^{m} k(x_i, x_j) - \frac{2}{mn} \sum_{i,j=1}^{m,n} k(x_i, y_j) + \frac{1}{n^2} \sum_{i,j=1}^{n} k(y_i, y_j). \tag{3.4}$$

One can also estimate the witness function from finite samples

$$\hat{f}(x) = \frac{1}{m} \sum_{i=1}^{m} k(x, x_i) - \frac{1}{n} \sum_{i=1}^{n} k(x, y_i) \tag{3.5}$$

i.e. the empirical witness function is the difference of two kernel density estimates [e.g. 17, 18]. This means that we can interpret the witness function as showing where the estimated densities of $p$ and $q$ are most different. While MMD two sample tests are well known in the literature the main contribution of this work is to show that this interpretability of the witness function makes them a useful tool as an exploratory form of statistical model criticism.

## 4 Examples on toy data

To illustrate the use of the MMD two sample test as a tool for model criticism we demonstrate its properties on two simple datasets and models.

**Newcomb's speed of light data** A histogram of Simon Newcomb's 66 measurements used to determine the speed of light [19] is shown on the left of figure 1. We fit a normal distribution to this data by maximum likelihood and ask whether this model is a faithful representation of the data.

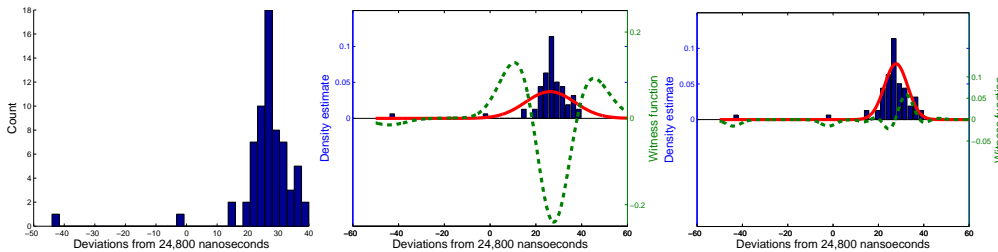

Figure 1: *Left*: Histogram of Simon Newcomb's speed of light measurements. *Middle*: Histogram together with density estimate (red solid line) and MMD witness function (green dashed line). *Right*: Histogram together with updated density estimate and witness function.

We sampled 1000 points from the fitted distribution and performed an MMD two sample test using a radial basis function kernel[2]. The estimated $p$-value of the test was less than $0.001$ i.e. a clear disparity between the model and data.

The data, fitted density estimate (normal distribution) and witness function are shown in the middle of figure 1. The witness function has a trough at the centre of the data and peaks either side indicating that the fitted model has placed too little mass in its centre and too much mass outside its centre.

This suggests that we should modify our model by either using a distribution with heavy tails or explicitly modelling the possibility of outliers. However, to demonstrate some of the properties of the MMD two sample test we make an unusual choice of fitting a Gaussian by maximum likelihood, but ignoring the two outliers in the data. The new fitted density estimate (the normal distribution) and witness function of an MMD test are shown on the right of figure 1. The estimated $p$-value associated with the MMD two sample test is roughly 0.5 despite the fitted model being a very poor explanation of the outliers.

The nature of an MMD test depends on the kernel defining the RKHS in equation (3.1). In this paper we use the radial basis function kernel which encodes for smooth functions with a typical lengthscale [e.g. 20]. Consequently the test identifies 'dense' discrepancies, only identifying outliers if the model and inference method are not robust to them. This is not a failure; a test that can identify too many types of discrepancy would have low statistical power (see [12] for discussion of the power of the MMD test and alternatives).

**High dimensional data**  The interpretability of the witness functions comes from being equal to the difference of two kernel density estimates. In high dimensional spaces, kernel density estimation is a very high variance procedure that can result in poor density estimates which destroy the interpretability of the method. In response, we consider using dimensionality reduction techniques before performing two sample tests.

We generated synthetic data from a mixture of 4 Gaussians and a $t$-distribution in 10 dimensions[3]. We then fit a mixture of 5 Gaussians and performed an MMD two sample test. We reduced the dimensionality of the data using principal component analysis (PCA), selecting the first two principal components. To ensure that the MMD test remains well calibrated we include the PCA dimensionality reduction within the bootstrap estimation of the null distribution. The data and plug-in predictive samples are plotted on the left of figure 2. While we can see that one cluster is different from the rest, it is difficult to assess by eye if these distributions are different — due in part to the difficulty of plotting two sets of samples on top of each other.

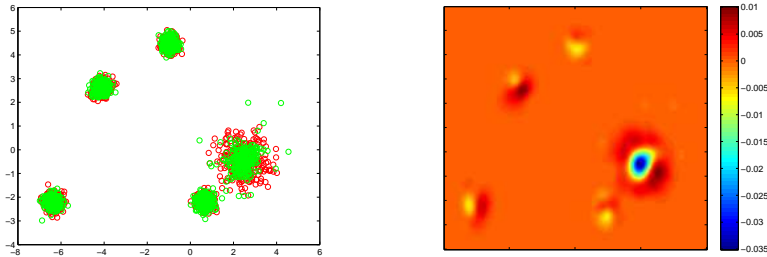

Figure 2: *Left*: PCA projection of synthetic high dimensional cluster data (green circles) and projection of samples from fitted model (red circles). *Right*: Witness function of MMD model criticism. The poorly fit cluster is clearly identified.

The MMD test returns a $p$-value of 0.05 and the witness function (right of figure 2) clearly identifies the cluster that has been incorrectly modelled. Presented with this discrepancy a statistical modeller might try a more flexible clustering model [e.g. 21, 22]. The $p$-value of the MMD statistic can also be made non-significant by fitting a mixture of 10 Gaussians; this is a sufficient approximation to the $t$-distribution such that no discrepancy can be detected with the amount of data available.

## 5   What exactly do neural networks dream about?

"To recognize shapes, first learn to generate images" quoth Hinton [23]. Restricted Boltzmann Machine (RBM) pretraining of neural networks was shown by [24] to learn a deep belief network (DBN) for the data i.e. a generative model. In agreement with this observation, as well as computing estimates of marginal likelihoods and testing errors, it is standard to demonstrate the effectiveness of a generative neural network by generating samples from the distribution it has learned.

When trained on the MNIST handwritten digit data, samples from RBMs (see figure 3a for random samples[4]) and DBNs certainly look like digits, but it is hard to detect any systematic anomalies purely by visual inspection. We now use MMD model criticism to investigate how faithfully RBMs and DBNs can capture the distribution over handwritten digits.

**RBMs can consistently mistake the identity of digits**   We trained an RBM with architecture $(784) \leftrightarrow (500) \leftrightarrow (10)$[5] using 15 epochs of persistent contrastive divergence (PCD-15), a batch size of 20 and a learning rate of 0.1 (i.e. we used the same settings as the code available at the deep learning tutorial [25]). We generated 3000 independent samples from the learned generative model by initialising the network with a random training image and performing 1000 gibbs updates with the digit labels clamped[6] to generate each image (as in e.g. [23]).

Since we generated digits from the class conditional distributions we compare each class separately. Rather than show plots of the witness function for each digit we summarise the witness function by examples of digits closest to the peaks and troughs of the witness function (the witness function estimate is differentiable so we can find the peaks and troughs by gradient based optimisation). We apply MMD model criticism to each class conditional distribution, using PCA to reduce to 2 dimensions as in section 4.

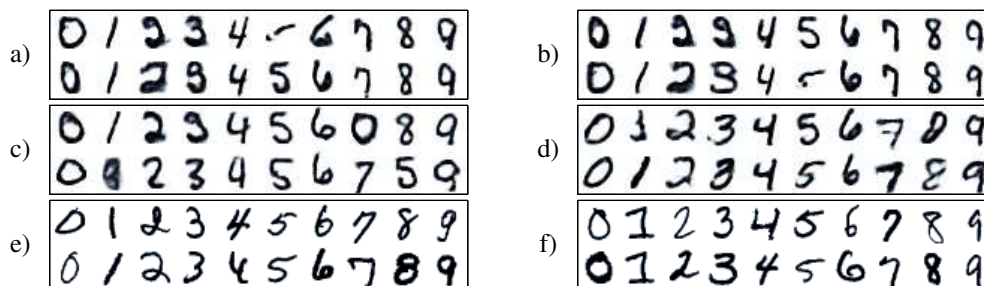

Figure 3: a) Random samples from an RBM. b) Peaks of the witness function for the RBM (digits that are over-represented by the model). c) Peaks of the witness function for samples from 1500 RBMs (with differently initialised pseudo random number generators during training). d) Peaks of the witness function for the DBN. e) Troughs (digits that are under-represented by the model) of the witness function for samples from 1500 RBMs. f) Troughs of the witness function for the DBN.

Figure 3b shows the digits closest to the two most extreme peaks of the witness function for each class; the peaks indicate where the fitted distribution over-represents the distribution of true digits. The estimated $p$-value for all tests was less than 0.001. The most obvious problem with these digits is that the first 2 and 3 look quite similar.

To test that this was not just an single unlucky RBM, we trained 1500 RBMs (with differently initialised pseudo random number generators) and generated one sample from each and performed the same tests. The estimated $p$-values were again all less than 0.001 and the summaries of the peaks of the witness function are shown in figure 3c. On the first toy data example we observed that the MMD statistic does not highlight outliers and therefore we can conclude that RBMs are making consistent mistakes e.g. generating a 0 from the 7 distribution or a 5 when it should have been generating an 8.

**DBNs have nightmares about ghosts**   We now test the effectiveness of deep learning to represent the distribution of MNIST digits. In particular, we fit a DBN with architecture $(784) \leftarrow (500) \leftarrow (500) \leftrightarrow (2000) \leftrightarrow (10)$ using RBM pre-training and a generative fine tuning algorithm described in [24]. Performing the same tests with 3000 samples results in estimated $p$-values of less than 0.001 except for the digit 4 (0.150) and digit 7 (0.010). Summaries of the witness function peaks are shown in figure 3d.

The witness function no longer shows any class label mistakes (except perhaps for the digit 1 which looks very peculiar) but the 2, 3, 7 and 8 appear 'ghosted' — the digits fade in and out. For comparison, figure 3f shows digits closest to the troughs of the witness function; there is no trace of ghosting. This discrepancy could be due to errors in the autoassociative memory of a DBN propogating down the hidden layers resulting in spurious features in several visible neurons.

# 6 An extension to non i.i.d. data

We now describe how the MMD statistic can be used for model criticism of non i.i.d. predictive distributions. In particular we construct a model criticism procedure for regression models.

We assume that our data consists of pairs of inputs and outputs $(x_i^{\mathrm{obs}}, y_i^{\mathrm{obs}})_{i=1\ldots n}$. A typical formulation of the problem of regression is to estimate the conditional distribution of the outputs given the inputs $p(y \,|\, x, \theta)$. Ignoring that our data are not i.i.d. we can generate data from the plug-in conditional distribution $y_i^{\mathrm{rep}} \sim p(y \,|\, x_i^{\mathrm{obs}}, \hat\theta)$ and compute the empirical MMD estimate (3.4) between $(x_i^{\mathrm{obs}}, y_i^{\mathrm{obs}})_{i=1\ldots n}$ and $(x_i^{\mathrm{obs}}, y_i^{\mathrm{rep}})_{i=1\ldots n}$. The only difference between this test and the MMD two sample test is that our data is generated from a conditional distribution, rather than being i.i.d. . The null distribution of this statistic can be trivially estimated by sampling several sets of replicate data from the plug-in predictive distribution.

To demonstrate this test we apply it to 4 regression algorithms and 13 time series analysed in [7]. In this work the authors compare several methods for constructing Gaussian process [e.g. 20] regression models. Example data sets are shown in figures 4 and 5. While it is clear that simple methods will fail to capture all of the structure in this data, it is not clear a priori how much better the more advanced methods will fair.

To construct $p$-values we use held out data using the same split of training and testing data as the interpolation experiment in [7][7]. Table 1 shows a table of $p$-values for 13 data sets and 4 regression methods. The four methods are linear regression (Lin), Gaussian process regression using a squared exponential kernel (SE), spectral mixture kernels [26] (SP) and the method proposed in [7] (ABCD). Values in bold indicate a positive discovery after a Benjamini–Hochberg [27] procedure with a false discovery rate of 0.05 applied to each model construction method.

| Dataset | Lin | SE | SP | ABCD |
|---|---|---|---|---|
| Airline | 0.34 | 0.36 | 0.07 | 0.15 |
| Solar | **0.00** | **0.00** | **0.00** | 0.05 |
| Mauna | **0.00** | 0.99 | 0.34 | 0.21 |
| Wheat | **0.00** | **0.00** | **0.00** | 0.19 |
| Temperature | 0.44 | 0.54 | 0.68 | 0.75 |
| Internet | **0.00** | **0.00** | 0.05 | **0.01** |
| Call centre | **0.00** | **0.02** | **0.00** | 0.07 |
| Radio | **0.00** | **0.00** | **0.00** | **0.00** |
| Gas production | **0.00** | **0.00** | **0.01** | 0.11 |
| Sulphuric | **0.00** | 0.29 | 0.34 | 0.52 |
| Unemployment | **0.00** | **0.00** | **0.00** | **0.01** |
| Births | **0.00** | **0.00** | **0.00** | 0.12 |
| Wages | **0.00** | **0.00** | **0.01** | **0.00** |

Table 1: Two sample test $p$-values applied to 13 time series and 4 regression algorithms. Bold values indicate a positive discovery using a Benjamini–Hochberg procedure with a false discovery rate of 0.05 for each method.

We now investigate the type of discrepancies found by this test by looking at the witness function (which can still be interpreted as the difference of kernel density estimates). Figure 4 shows the solar and gas production data sets, the posterior distribution of the SE fits to this data and the witness functions for the SE fit. The solar witness function has a clear narrow trough, indicating that the data is more dense than expected by the fitted model in this region. We can see that this has identified a region of low variability in the data i.e. it has identified local heteroscedasticity not captured by the model. Similar conclusions can be drawn about the gas production data and witness function.

Of the four methods compared here, only ABCD is able to model heteroscedasticity, explaining why it is the only method with a substantially different set of significant $p$-values. However, the procedure is still potentially failing to capture structure on four of the datasets.

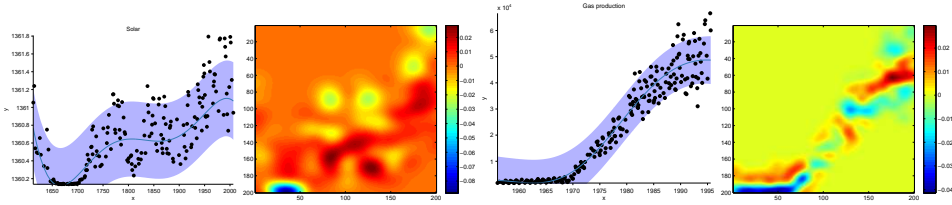

Figure 4: *From left to right*: Solar data with SE posterior. Witness function of SE fit to solar. Gas production data with SE posterior. Witness function of SE fit to gas production.

Figure 5 shows the unemployment and Internet data sets, the posterior distribution for the ABCD fits to the data and the witness functions of the ABCD fits. The ABCD method has captured much of the structure in these data sets, making it difficult to visually identify discrepancies between model and data. The witness function for unemployment shows peaks and troughs at similar values of the input $x$. Comparing to the raw data we see that at these input values there are consistent outliers. Since ABCD is based on Gaussianity assumptions these consistent outliers have caused the method to estimate a large variance in this region, when the true data is non-Gaussian. There is also a similar pattern of peaks and troughs on the Internet data suggesting that non-normality has again been detected. Indeed, the data appears to have a hard lower bound which is inconsistent with Gaussianity.

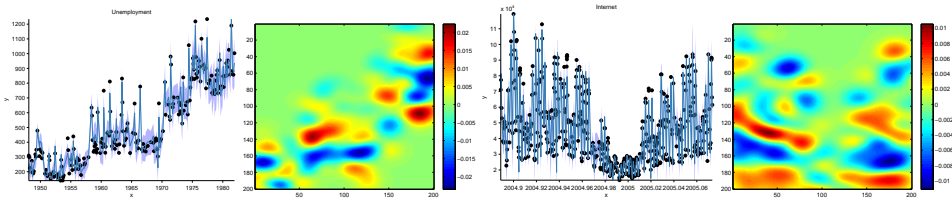

Figure 5: *From left to right*: Unemployment data with ABCD posterior. Witness function of ABCD fit to unemployment. Internet data with ABCD posterior. Witness function of ABCD fit to Internet.

## 7    Discussion of model criticism and related work

**Are we criticising a particular model, or class of models?**    In section 2 we interpreted the differences between classical, Bayesian prior/posterior and plug-in $p$-values as corresponding to different null hypotheses and interpretations of the word 'model'. In particular classical $p$-values test a null hypothesis that the data could have been generated by a class of distributions (e.g. all normal distributions) whereas all other $p$-values test a particular probability distribution.

Robins, van der Vaart & Ventura [28] demonstrated that Bayesian and plug-in $p$-values are not classical $p$-values (frequentist $p$-values in their terminology) i.e. they do not have a uniform distribution under the relevant null hypothesis. However, this was presented as a failure of these methods; in particular they demonstrated that methods proposed by Bayarri & Berger [29] based on posterior predictive $p$-values are asymptotically classical $p$-values.

This claimed inadequacy of posterior predictive $p$-values was rebutted [30] and while their usefulness is becoming more accepted (see e.g. introduction of [31]) it would appear there is still confusion on the subject [32]. We hope that our interpretation of the differences between these methods as different null hypotheses — appropriate in different circumstances — sheds further light on the matter.

**Should we worry about using the same data for traning and criticism?**    Plug-in and posterior predictive $p$-values test the null hypothesis that the observed data could have been generated by the fitted model or posterior predictive distribution. In some situations it may be more appropriate to attempt to falsify the null hypothesis that future data will be generated by the plug-in or posterior predictive distribution. As mentioned in section 2 this can be achieved by reserving a portion of the data to be used for model criticism alone, rather than fitting a model or updating a posterior on the full data. Cross validation methods have also been investigated in this context [e.g. 33, 34].

**Other methods for evaluating statistical models** Other typical methods of model evaluation include estimating the predictive performance of the model, analyses of sensitivities to modelling parameters / priors, graphical tests, and estimates of model utility. For a recent survey of Bayesian methods for model assessment, selection and comparison see [35] which phrases many techniques as estimates of the utility of a model. For some discussion of sensitivity analysis and graphical model comparison see [e.g. 4].

In this manuscript we have focused on methods that compare statistics of data with predictive distributions, ignoring parameters of the model. The discrepancy measures of [36] compute statistics of data and parameters; examples can be found in [4]. O'Hagan [2] also proposes a method and selectively reviews techniques for model criticism that also take model parameters into account.

In the spirit of scientific falsification [e.g. 37], ideally all methods of assessing a model should be performed to gain confidence in any conclusions made. Of course, when performing multiple hypothesis tests care must be taken in the intrepetation of individual $p$-values.

## 8 Conclusions and future work

In this paper we have demonstrated an exploratory form of model criticism based on two sample tests using kernel maximum mean discrepancy. In contrast to other methods for model criticism, the test analytically maximises over a broad class of statistics, automatically identifying the statistic which most demonstrates the discrepancy between the model and data. We demonstrated how this method of model criticism can be applied to neural networks and Gaussian process regression and demonstrated the ways in which these models were misrepresenting the data they were trained on.

We have demonstrated an application of MMD two sample tests to model criticism, but they can also be applied to any aspect of statistical modelling where two sample tests are appropriate. This includes for example, Geweke's tests of markov chain posterior sampler validity [38] and tests of markov chain convergence [e.g. 39].

The two sample tests proposed in this paper naturally apply to i.i.d. data and models, but model criticism techniques should of course apply to models with other symmetries (e.g. exchangeable data, logitudinal data / time series, graphs, and many others). We have demonstrated an adaptation of the MMD test to regression models but investigating extensions to a greater number of model classes would be a profitable area for future study.

We conclude with a question. Do you know how the model you are currently working with most misrepresents the data it is attempting to model? In proposing a new method of model criticism we hope we have also exposed the reader unfamiliar with model criticism to its utility in diagnosing potential inadequacies of a model.

## Footnotes

[1]We follow Box [1] using the term 'model criticism' for similar reasons to O'Hagan [2].

[2] Throughout this paper we estimate the null distribution of the MMD statistic using the bootstrap method described in [12] using 1000 replicates. We use a radial basis function kernel and select the lengthscale by 5 fold cross validation using predictive likelihood of the kernel density estimate as the selection criterion.

[3]For details see code at [redacted]

[4] Specifically these are the activations of the visible units before sampling sampling binary values. This procedure is an attempt to be consistent with the grayscale input distribution of the images. Analogous discrepancies would be discovered if we had instead sampled binary pixel values.

[5]That is, 784 input pixels and 10 indicators of the class label are connected to 500 hidden neurons.

[6]Without clamping the label neurons, the generative distribution is heavily biased towards certain digits.

[7]Gaussian processes when applied to regression problems learn a joint distribution of all output values. However this joint distribution information is rarely used; typically only the pointwise conditional distributions $p(y \,|\, x_i^{\mathrm{obs}}, \hat\theta)$ are used as we have done here.

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
