[Reviews · NeurIPS 2015]

Submitted by Assigned_Reviewer_1

Summary:

This paper describes an approach to statistical model criticism using the kernel two-sample test maximum mean discrepancy. The idea behind model criticism is simply to assess the ability of a given model to explain the observed data, and more importantly, to determine in which regions of the space the data is most misinterpreted by the model. For this purpose, the witness function of the MMD test is employed. This function takes large absolute values where the predictive distribution of the model considered is most different from the distribution of the actual observed data. The benefits of the approach described are shown in experiments involving restricted Boltzmann machines, deep networks and Gaussian processes.

This paper covers in detail the task of model criticism in which one seeks the aspects of the observed data that the particular model cannot explain very well. This is an important tasks that may have been over-looked by the machine learning community. The main problem is however that, besides this, there is nothing really new in this paper. The methods it is based on (the MMD tests) are very well known and established. Something that would have been very interesting is to propose, besides the use of the witness function, something actually new to identify the misinterpretations of the model. In particular, in the experiments in Section 5, where one cannot simply plot the witness function, I miss a bit this aspect.

Quality:

The quality of the paper is high. It describes in detail the approaches considered by the authors and the paper contains illustrative experiments of the methods described including several datasets and models. Thus, I believe this paper is of good quality. Some illustrative figures that show the value of the witness function, which indicates the regions of mismatch between the model and the observed data are also included in the paper.

Clarity:

Overall the paper is very well written with detailed explanations. Thus I believe the clarity of the paper is high. Unfortunately, there is a section in which the approach described and the results obtained is not very clear. This section is Section 5. This is probably due to the fact that is not possible to plot the witness function for the MNIST dataset considered there.

Originality:

The use of the MMD test for model criticism seems to be novel. However, besides that, there not much novelty in the proposed paper. In particular the MMD test has been well known within the machine learning community since several years, and this paper uses it as such. The utility of the witness function to depict differences between both probability distributions was also known.

This may be the only weak point of the paper.

Significance:

I believe this paper may be significant. In particular, it shows concerns about particular aspects that machine learners typically ignore. Namely, the identification of the weakest points of a particular model chosen to explain the observed data. The identification of these points may rise concerns about the particular choice of the model, and may be very useful to modify or update the model in consequence to better explain the observed data.
Summary: Very interesting paper that proposes a novel use of the MMD statistical test, which may be interesting for the machine learning community, and nothing else.

Submitted by Assigned_Reviewer_2

This paper suggests that MMD is good test validate if statistical models learn the true distribution of the data.

- no theoretical contribution, but very nice use of existing theory and techniques. - nice empirical contribution (nightmares etc.)

- showing empirically that MMD ignores outliers and so mistakes in RBM are not of the outliers type (how it works in theory ?). - simple and effective idea to find peaks of the witness function. -'We now describe how the MMD statistic can be used for model criticism of non i.i.d. predictive distributions' - you could have used

MMD for non iid data.

- I am not entirely sure that 'The null distribution of this statistic can be trivially estimated by sampling several sets of replicate data from the plug-in predictive distribution'. I think that the errors from the plug-in predictive conditional distribution might change the null distribution. Besides I don't know if I understand the procedure. - in cited paper 7 they have time series (I had a look at it and some of which might not be stationary, so one needs to be extra careful with two-sample test) -- again for time series I suggest you run KPSS test,

Dickey-Fuller test decide if time-series seems to be stationary and then run wild-bootstrap MMD. Such a procedure is more statistically sound, I think.

Summary: Not exactly very

theoretically original, but very nice use of existing theory and original empirically.

Submitted by Assigned_Reviewer_3

This paper proposes a procedure for using the MMD, a kernel two-sample test, as an exploratory tool for model criticism. The main contribution of this paper is a novel interpretation of the MMD witness function, which results in a procedure for model criticism that maximizes over a broad class of measures of discrepancy between distributions and automatically identifies regions of the data which are most poorly represented by the model. The authors demonstrate their approach on i.i.d. and time-series data using toy examples, RBMs, and GPR and include significant interesting discussions.

While there is not a considerable amount of new technical work, the author's interpretation of the MMD and the framework propose is novel and has advantages over common existing approaches to model criticism. Since model criticism is important for anyone working in machine learning this could have broach impact in the community. The paper is well written and very easy to understand and the applications are well presented with considerable attention to detail.
Summary: This paper proposes using the MMD, a kernel two-sample test, for model criticism. The authors' interpretation of the MMD witness function and framework for model criticism are novel and have advantages over common approaches for model criticism. The authors present their method very clearly and include well presented applications.

Submitted by Assigned_Reviewer_4

The function that attains the supremum of the statistic used in MMD two-sample tests can be used to identify regions in attribute space in which a statistical model most misrepresents the data used for automatic induction. This "witness function" can be interpreted as the difference of two kernel density estimates one corresponding to the actual data and one to the data generated with the fitted model. The authors then apply the procedure to real and synthetic learning problems and illustrate (rather than demonste, as the authors claim) the ways in which automatically induced models, such as restricted Boltzmann machines, deep belief networks or Gaussian processes regressors fail to capture the properties of the data on which they are trained.

The paper is very well written and is a pleasure to read. However the originality and significance of the contribution is rather limited. Part of the problem stems from the lack of a clear focus: it is presented as a "an exploratory approach" that touches on many aspects of Statistical Model Criticism but fails to provide sufficient background or depth in the discussion of the issues addressed. This would be resolved in a longer version that provides a more detailed discussion.

According to the authors, the main contribution of this work is highlighting the importance for model criticism of the (known) property that the witness function in MMD two-sample tests is the difference of two kernel density estimates. This releveance apparently stems from the fact that the MMD statistic is the supremum among a large space o possible statistics.

Leaving aside the connection with MMD, the witness function is simply the difference of kernel density estimates. If one were asked to identify regions in which a statistical model most misrepresents the data it was trained on, this method is probably the most obvious choice.

In criticising a model one should have the use of the model in mind. Therefore, model evaluation ought to be guided by a utility function. I agree with the authors that focusing on the differences of kernel density estimates for the actual data and for data generated by the fitted model is probably an excellent general purpose model criticism tool, especially if I am interested in local differences. I would argue (rather trivially) that the important feature is the local character of RBF kernels. The maximization property is reassuring, albeit not of much practical use:

The crucial issue seems to be the choice of kernel and kernel parameters. Unfortunately, in MMD two-sample tests, these need to be specified by the user.

In summary, the paper is based on a fairly straightforward result: that differences of kernel densitie estimates help us understand how and where a model fails to fit the data. The experimental investigation is promising but seems to be rather preliminary and unsystematic.

On the question posed by the authors: would it be worth this paper being published now as a conference paper to bring model criticism to the attention of the machine learning community quickly or waiting a year or two for a review article?

Yes, it would be worth waiting.

A minor comment:

"While it is clear that simple methods will fail to capture all of the structure in this data, it is not clear a priori how much better the more advanced methods will fair."

In line 297, the authors probably meant "will fare".

Summary: This is an erudite, well-written paper, that brings Model Criticism to the attention of the machine learning community. Unfortunately, the essay fails to fulfill its lofty albeit poorly defined goal. The originality of the contribution of the research is rather limited. The treatment is insufficient either as a tutorial or as a survey in the area. I believe that the emphasis on the fact that the kernel density estimation on which the model criticism analysis is based is used to compute the witness function in MMD two-sample tests is not particularly relevant to the subsequent discussion. Nevertheless, Statistical Model Criticism probably deserves more attention in Machine learning. I would suggest that the authors elaborate and improve the focus of this "exploratory approach" and publish it as a tutorial or a review in the area.

Author Feedback
Author rebuttal: We would first like to thank all of the reviewers for their time and insights. All of the reviewers have understood this paper; it is a novel application of existing techniques to an important but under-investigated area. The only truly negative review is that of reviewer 4 suggesting that we write a review article or similar since "Model Criticism probably deserves more attention in Machine learning" and they consider the current article to be an insufficient treatment. However, reviewer 4 also says, "This is an erudite, well-written paper, that brings Model Criticism to the attention of the machine learning community". Our question to reviewer 4 is thus, would it be worth this paper being published now as a conference paper to bring model criticism to the attention of the machine learning community quickly or waiting a year or two for a review article? The acceptance/rejection of this paper probably hinges on your review score.

Specific replies to queries / suggestions:

Reviewer 2 - non iid data. You are quite correct that we should cite the wild bootstrap work - we will add this. However, as far as we understand, the wild bootstrap work in its current form would not lend itself to the identification of the location of discrepancies, since it does not produce a witness function like object as a by product. However, it would be interesting in further work to see if the test can be understood as identifying locations of discrepancy.

Many reviewers - kernel choice. We agree with the reviewers that we should discuss kernel choice more thoroughly. Identification of some 'optimal' kernel choice is probably beyond the scope of this paper, but we can certainly elaborate on the effect of the choice. In particular, we can expand our remark that the test based on an RBF kernel is insensitive to outliers.